# Non-Invasive Fluorescent Monitoring of Ovarian Cancer in an Immunocompetent Mouse Model

**DOI:** 10.3390/cancers11010032

**Published:** 2018-12-31

**Authors:** Amy L. Wilson, Kirsty L. Wilson, Maree Bilandzic, Laura R. Moffitt, Ming Makanji, Mark D. Gorrell, Martin K. Oehler, Adam Rainczuk, Andrew N. Stephens, Magdalena Plebanski

**Affiliations:** 1Hudson Institute of Medical Research, Clayton 3168, Australia; Amy.Wilson@monash.edu.au (A.L.W.) Maree.Bilandzic@Hudson.org.au (M.B.); Laura.Moffitt@monash.edu.au (L.R.M.); Ming.Makanji@Hudson.org.au (M.M.); Adam.Rainczuk@Hudson.org.au (A.R.); 2Department of Molecular and Translational Sciences, Monash University, Clayton 3168, Australia; 3Department of Immunology and Pathology, Monash University, Clayton 3168, Australia; Kirsty.Wilson@monash.edu.au; 4School of Health and Biomedical Sciences, RMIT University, Bundoora 3083, Australia; 5Centenary Institute, The University of Sydney, Sydney 2006, Australia; m.gorrell@centenary.org.au; 6Department of Gynaecological Oncology, Royal Adelaide Hospital, Adelaide 5000, Australia; martin.oehler@adelaide.edu.au; 7Robinson Institute, University of Adelaide, Adelaide 5000, Australia; 8Bruker Biosciences Pty Ltd., Preston 3072, Australia

**Keywords:** iRFP, iRFP720, ovarian cancer, ID8, syngeneic, immune, T cell, tumour

## Abstract

Ovarian cancers (OCs) are the most lethal gynaecological malignancy, with high levels of relapse and acquired chemo-resistance. Whilst the tumour–immune nexus controls both cancer progression and regression, the lack of an appropriate system to accurately model tumour stage and immune status has hampered the validation of clinically relevant immunotherapies and therapeutic vaccines to date. To address this need, we stably integrated the near-infrared phytochrome iRFP720 at the *ROSA26* genomic locus of ID8 mouse OC cells. Intrabursal ovarian implantation into C57BL/6 mice, followed by regular, non-invasive fluorescence imaging, permitted the direct visualization of tumour mass and distribution over the course of progression. Four distinct phases of tumour growth and dissemination were detectable over time that closely mimicked clinical OC progression. Progression-related changes in immune cells also paralleled typical immune profiles observed in human OCs. Specifically, we observed changes in both the CD8+ T cell effector (Teff):regulatory (Treg) ratio, as well as the dendritic cell (DC)-to-myeloid derived suppressor cell (MDSC) ratio over time across multiple immune cell compartments and in peritoneal ascites. Importantly, iRFP720 expression had no detectible influence over immune profiles. This new model permits non-invasive, longitudinal tumour monitoring whilst preserving host–tumour immune interactions, and allows for the pre-clinical assessment of immune profiles throughout disease progression as well as the direct visualization of therapeutic responses. This simple fluorescence-based approach provides a useful new tool for the validation of novel immuno-therapeutics against OC.

## 1. Introduction

Ovarian cancer (OC) is the most lethal gynaecological malignancies, typically characterised by asymptomatic progression, diagnosis at an advanced stage, and a high rate of recurrence [1]. The majority of OC patients ultimately develop platinum-resistant disease, limiting therapeutic options and underlying the very high mortality rate from this disease. Therapeutic vaccines based on antigenic protein epitopes, DNA or pulsed dendritic cells (DCs) offer one promising approach to stimulating immune activation in vivo [2,3]. Vaccination has proven successful in other gynaecological tumour types (e.g., cervical cancers), and numerous antigenic targets (e.g., cancer–testis antigen sperm protein 17 (SP17) [4]) have been suggested in OC. However, immunotherapy trials in OC (e.g., immune checkpoint inhibitors, and therapeutic vaccines) have failed in inducing the clinically robust and durable responses achieved for other tumour types (reviewed in [5]). There remains a clear need for improved therapies that can regress disease against the background of chemo-resistance and immune suppression endemic to these cancer types.

The lack of effective immune-targeted OC therapies to date suggests that improved pre-clinical in vivo models are required to model therapeutic response and accurately assess efficacy. Orthotopic patient-derived xenograft (PDX) models necessarily lack the key tumour–immune interactions that drive tumour growth and progression [6], whilst no available genetically engineered mouse models (GEMMs) accurately recapitulate all features of human disease (reviewed in [7,8]). Syngeneic models (including ID8 [9], MOSE-L [10], STOSE [11]) offer an excellent alternative for the study of OC as they are generally well characterised in vitro, readily amenable to ex vivo genetic manipulation, and facilitate the reliable, rapid development of tumours following implant. Of these, the ID8 model is the most extensively studied in the context of cancer biology and tumour–host immune interaction [12,13], making it the model of choice for analyses of antigenic determinants in vivo [4]. The limitations of this model, however, include slow development, uncertainty around the molecular mechanisms driving their malignant potential [14], and an inability to accurately monitor real-time tumour progression in the context of human disease [15]. 

Tumour–immune interactions are dynamic, changing over time and in response to therapy [16]. OC models must therefore replicate the complex tumour microenvironment, permit quantitative detection and analysis of low-volume tumours, allow for high-fidelity monitoring of tumour progression over time, and permit correlations between tumour growth, spread and changes in immune response. Bioluminescence imaging using luciferin injection into mice bearing ID8 *luc*+ cells has previously been used to image ID8 tumours during progression [17]. However, luciferin injection is invasive and can initiate an inflammatory response, and only provides qualitative information regarding tumour progression [18]. Alternatives such as green fluorescent proteins (GFPs) are widely used in vitro for diverse applications, but their in vivo application is unfavourable due to extensive tissue absorption and autofluorescence [19,20]. The more recent development of near-infrared fluorescent proteins (iRFPs) for non-invasive deep tissue imaging could potentially overcome these drawbacks [21]. Tissue absorption in the near-infrared spectrum is minimal, and the relatively low level of autofluorescence and light scatter by mammalian tissues at these wavelengths make iRFPs desirable for deep tissue imaging [21]. Near-infrared fluorescent proteins have been used to track the progression of prostate cancer [21], brain tumours [22], lung adenocarcinoma [23], glioblastoma, osteosarcoma, and melanoma [24], but have not been applied in the context of OC.

In this study, we have developed and characterized a new ID8 tumour model, utilizing a near infra-red fluorescent protein (iRFP720 [21]) stably integrated into the *ROSA26* locus of ID8 cells to non-invasively monitor and directly stage tumour progression in mice. This approach permits real-time imaging of tumour progression and evaluation of response to therapy and particularly stage-specific therapeutic vaccines, enabling the direct analysis of immune-mediated tumour regression in vivo.

## 2. Results

### 2.1. Generation of Clonal ID8 Cells Expressing Genomically Integrated iRFP720

The pROSA26-puro-iRFP720 vector (Figure 1) was constructed to constitutively express iRFP720 from a cytomegalovirus (CMV) promoter, puromycin and ampicillin selection markers, and flanking homology arms corresponding to upstream and downstream regions of the *ROSA26* locus. Successful genomic integration transfers both CMV-iRFP720 and PGK-puro cassettes, conferring puromycin resistance and stable expression of iRFP720. ID8 cells were co-transfected with pROSA-puro-iRFP720 and plentiCRISPRv1 vectors, containing either ROSA26-sgRNA2 or ROSA26-sgRNA4 [25]. Following single cell sorting for iRFP720+ cells, clonal expansion and fluorescence microscopy were used to confirm iRFP+ status (Figure 2A,B). Individual clones were then screened by PCR for genomic integration of iRFP720 into *ROSA26* (Figure 2C,D). Screening primers were directed against: (i) an endogenous ROSA26 region upstream of the incorporation; (ii) an endogenous ROSA26 region between the left and right arms; and (iii) a region specific to the iRFP720 sequence (as depicted in Figure 2C). This approach differentiates between homo- and heterogeneous clones. Heterozygous incorporation of iRFP720 into the *ROSA26* locus was confirmed in 40% of clones (Figure 2D); no clones homozygous for iRFP720 integration were detected. A single iRFP+ clone was selected for all subsequent studies.

### 2.2. Expression of iRFP720 Does Not Alter ID8 Cell Growth Characteristics or Chemo-Sensitivity In Vitro

Following genomic incorporation of iRFP720 into ID8 cells, we assessed both their proliferative capacity and response to chemotherapy compared to wild-type ID8 cells in vitro. Using xCELLigence real-time cell analysis (RTCA), we observed no significant difference in the adhesive or proliferative capacity of pROSA-iRFP720 ID8 cells compared to wild-type ID8 cells over 72 h (Figure 2E). Similarly, there was no significant difference in cell index (CI) between wild-type or iRFP+ ID8 cells following treatment with cisplatin (10 µg/mL) or paclitaxel (20 nM) (Figure 2E). We also assessed apoptosis 24 h post-chemotherapy by Annexin V and propidium iodide staining. Again, pROSA-iRFP720 ID8 cells displayed no apparent difference in apoptotic cell death following cisplatin or paclitaxel treatment compared to wild-type (Figure 2F,G). These data demonstrate that the genomic incorporation and subsequent constitutive expression of iRFP720 had no influence over ID8 cell growth, proliferation, chemo-sensitivity or apoptosis in vitro.

### 2.3. Non-Invasive Detection and Imaging of iRFP720 Fluorescence In Vivo

To determine the minimum cell titre for detection of iRFP720+ ID8 cells in vivo, cells were injected intrabursally (from 6.25 × 10^4^ to 1 × 10^6^ cells/ovary) and mice were imaged using an IVIS Lumina III Imaging System. A minimum titre of 0.5 × 10^6^ cells was required for detection of iRFP+ ID8 cells following implant, with robust detection at 1 × 10^6^ cells (data not shown). We therefore used 1 × 10^6^ cells for all subsequent intrabursal implant experiments.

Mice implanted with 1 × 10^6^ iRFP720+ ID8 cells were monitored over a period of 10 weeks (humane endpoint). All mice were fed a low-fluorescence chow diet to minimize potential autofluorescence; and a spectral unmixing algorithm was created for imaging, to assist in visualization of iRFP+ signal (see Section 4.9). Fluorescence associated with implanted cancer cells was clearly detected immediately following implant, and increased over time corresponding to the growth of iRFP+ tumours (Figure 3A). 

### 2.4. iRFP720 Fluorescence Corresponds with Discrete Tumour Progression Stages In Vivo

To evaluate whether iRFP720 could be used to monitor tumour growth over time, mice implanted with iRFP720+ ID8 were imaged weekly for fluorescence. At 2-week intervals (and at the endpoint), mice were culled and an autopsy was performed to assess primary tumour size, localization and number of metastatic nodules. Weight and abdominal girth, the most commonly used surrogate parameters of tumour progression, were also measured and correlated with iRFP fluorescence and macroscopic tumour deposition.

Both the intensity and distribution of iRFP fluorescence increased over time in iRFP+ ID8 mice compared to those in non-fluorescent controls (Figure 3A,B), and displayed a biphasic pattern with two distinct fluorescence peak and trough events (Figure 3B). Correlation of these fluorescent events with tumour growth and spread at autopsy allowed us to define four distinct stages in ID8 tumour progression. At fluorescence peak event 1 (3 weeks post-implant), mice had small primary tumours that remained confined to the implant site, with some peritoneal adhesions but no evidence of capsule disruption or dissemination of cancer cells. Accordingly, this was defined as a Stage I tumour. Immediately following stage I, fluorescence decreased; in addition to primary tumours and extensive local adhesions to the peritoneal wall, at this time we observed disruption of the ovarian capsule and the first evidence of micro-metastases (observed by iRFP fluorescence) within the peritoneum. This initial decrease event was therefore defined as the transition to Stage II. Fluorescence then steadily increased to reach a second maxima after 8 weeks by which point mice had developed large primary tumours, extensive peritoneal adhesions and multiple metastatic nodules throughout the peritoneal cavity involving the peritoneum, small intestine, liver, kidneys and omentum. This second fluorescence maxima was defined as Stage III and was again followed by a decline in fluorescence corresponding to the development of peritoneal ascites. At the endpoint, mice had developed large primary tumours, extensively disseminated secondary tumours, and accumulated peritoneal ascites. By contrast to fluorescence measurements, weight gain proceeded in a linear fashion over time whilst abdominal circumference only evidenced an increase from ~week 7 onwards (Figure 3C). 

Tumour tissues at the endpoint were also evaluated for iRFP fluorescence ex vivo, and immuno-stained for common markers of serous cancers including pan-cytokeratin (pan-CK), programmed death-ligand 1 (PD-L1), p53 and Wilms tumour protein (WT1). Consistent with previous studies [26], tumour deposition at the endpoint was present on intestines, peritoneum, liver and stomach; and iRFP720 fluorescence was evident in all tumour deposits (Figure 3D). The expression of all tumour markers was similar between wild-type and pROSA-iRFP720 ID8 tumour tissues (Figure 4A–E), indicating that iRFP720 expression did not alter their abundance.

Real-time monitoring using iRFP was thus able to define discrete tumour stages that corresponded closely to the clinical progression of OC. Moreover, fluorescence measurements permitted the detection of tumour progression at a significantly earlier time-point than the commonly used measurements of weight and abdominal girth. Taken together, our data indicate that iRFP fluorescence is suitable for non-invasive monitoring of stage-specific tumour progression in vivo.

### 2.5. The Presence of Anti-iRFP720 Antibodies Does Not Impact Tumour Progression

Expression of a foreign protein can stimulate an autoimmune response [27], to potentially alter tumour progression or overall survival [17]. ELISA was used to test for the presence of circulating IgG against the iRFP720 protein. Only 2/14 mice (14%) had a detectible titre (~1 µg/mL) of anti-iRFP720 IgG (Figure 5), and there was no discernible effect on disease progression in either of these animals or for iRFP+ ID8 tumour mice generally compared to that in wild-type cells. This is consistent with findings for mice implanted with ID8-*luc* cells [17]. Thus, iRFP720 was only weakly immunogenic in C57BL/6 mice and its expression did not alter tumour progression in vivo.

### 2.6. iRFP720 Expression Does Not Alter the Tumour Immune Microenvironment In Vivo 

To achieve utility as a model for tumour immune studies, we performed extensive leukocyte profiling to establish a baseline for tumour–immune interactions and further confirm the equivalence of iRFP720 ID8 mice to wild-type ones. Mice were culled at weeks 4 and 8 post-inoculation, corresponding to early dissemination (stage 2) or advanced disease (stage 3), and leukocyte populations present in blood, peritoneal fluid and lymphoid organs (nodes, spleen) were evaluated by flow cytometry. No significant changes between iRFP+ vs. wild-type mice were identified in any of the macrophage, MDSC, DC or multiple T-cell populations examined (Figure 6). Between “early” (stage II) and “late” (stage III) tumour dissemination, we observed that percentages of total CD3+ T cells in all tissues examined were largely unchanged over time. However, a significant increase in Treg cells (CD3+CD4+CD25+FoxP3+) occurred in all tissues tested (Figure 6B), and a corresponding decrease in Teff cells (CD3+CD8+) in lymph nodes (Figure 6C) led to an overall decrease in the Teff:Treg ratio in the blood and peritoneum (Figure 6D). Macrophages (CD3-CD11b+F4/80+MHCII+) residing in the peritoneal cavity and lymphoid organs also decreased over time (Figure 6E), as did MDSCs (CD3−CD11c−D11b+GR1+) in blood and lymphoid tissues; but in the peritoneal cavity, there was an ~3-fold increase in MDSCs (Figure 6F). The percentage of DCs (CD3−GR1−CD11c+) decreased in lymphoid organs, but were unchanged in the blood or peritoneum (Figure 6G). These data are largely consistent with the well-characterised immune profile of human OCs [28], and demonstrate that iRFP expression did not influence the overall immune profile. Thus, our data show that iRFP720 ID8 cells are a suitable model for ongoing studies of tumour–immune interactions in OC.

## 3. Discussion

Immune-mediated therapies have been successful in many solid tumour types, and several checkpoint inhibitors and therapeutic vaccines are now FDA-approved for cancer-related indications [29,30,31,32]. However, none has been successful in clinical trials for OC to date. The majority of pre-clinical models rely on xenografted tissue explants; these models do not recapitulate either the immune-mediated drivers of tumour progression or the complexity of the anti-tumour immune response [33,34]. Moreover, dynamic changes in the tumour microenvironment drive, and in turn are driven by immune-mediated events over time and according to tumour progression [16,35]. It is therefore crucial to develop an appropriate pre-clinical model that permits the evaluation of tumour dynamics, allows for direct correlation with tumour stage, and operates in the context of an immune-competent microenvironment. Such a model is essential to appropriately assess and validate immune-directed therapies for successful translation in OC.

We have demonstrated that the near-infrared fluorescent protein iRFP720, a bacterial phytochrome with far-red spectral emission [21], can be stably integrated into the genome of ID8 OC cells and can be used to non-invasively track tumour progression over time. iRFPs have been used for deep tissue imaging in several systems [22,23,24,36], but have never been applied in the context of OC. Integration of a constitutively expressed iRFP720 at the *ROSA26* genomic locus [37] had no discernible consequences for cell growth in vitro, nor did it induce any significant changes in either cellular or humoral immune responses in vivo. Moreover, low-titre anti-iRFP antibodies could only be detected in 2/14 mice with advanced disease; and iRFP expression had no discernible impact on tumour growth, progression or survival time in any animals. Whilst other fluorescent proteins (e.g., enhanced green fluorescent protein) can induce an autoantibody response with subsequent effects on tumour growth [38], our data are similar to a previous report utilizing randomly integrated *Renilla spp.* luciferase in ID8 cells where no significant immune response was detected [17]. Moreover, CRISPR-mediated integration of the iRFP cassette at the “safe harbour” *ROSA26* locus, extensively validated for GEMM construction [37], was utilized to ensure minimal potential for genomic disruption. Our data demonstrate that iRFP fluorescence is an effective tool for monitoring tumour-specific changes in vivo, has low immunogenicity and does not exert any major alterations to cellular phenotype.

Fluorescence imaging allowed us to define four distinct stages of tumour progression in the ID8 model, which closely mimic the progression of human OCs [39]. Discrete fluorescence events were directly correlated with primary tumour growth (Stage I), early dissemination (Stage II), advanced dissemination (Stage III) and the accumulation of ascites fluid. Importantly, these events were identified simply and non-invasively permitting the direct monitoring of tumour growth. Fluorescence monitoring was far superior to the standard surrogate measurement of abdominal girth, where changes did not become evident until week 7, at which point mice had already progressed to stage III disease. Unlike luciferase-based imaging, which can induce inflammation and permits only qualitative assessment of tumour presence/absence [18,19], fluorescence monitoring does not require an endogenous substrate and can be assessed quantitatively. The ability to make repeated measurements during progression, monitor response to therapy and directly correlate changes according to tumour stage adds a powerful new dimension to the ID8 OC model.

Immune-mediated destruction of OC is a multi-faceted process, requiring the co-ordination of multiple cell types, their activation status and the complex interplay to balance immune active vs. suppressive phenotypes. OC is generally characterised by a decreased tumour-infiltrating CD8+/T regulatory cell ratio, accumulation of MDSCs), and a high MDSC-to-DC ratio [40,41,42,43,44]. The consequent accumulation of MDSCs and increased ratios of infiltrating T-regulatory to T-effector (Treg:Teff) or ascites-associated MDSC:DCs are thus independent prognostic predictors for OC patients [41,44]. Each of these features was paralleled by the ID8-iRFP720 model, and accumulated progressively from early metastatic to late metastatic disease. This demonstrates not only the suitability of the ID8-iRFP720 model for the stage-specific analysis of immune cell changes, but also the potential to monitor complex changes beyond simple alterations in cell numbers. For example, the Teff cell generation, tumour trafficking and infiltration, activation status, avidity for tumour antigens, longevity and differentiation state are all highly variable and expected to alter throughout the course of disease. By staging and monitoring these events longitudinally throughout progression, the ID8-iRFP720 model has the potential to unparalleled depth for the investigation of tumour–immune interplay over time. 

There is increasing interest in the development of therapeutic vaccines for OC treatment, aimed at inducing or altering specific aspects of immunity to promote immune-mediated tumour regression [2,45,46,47]. For example, vaccination of mice with irradiated mouse ovarian surface epithelial cells (MOSEC) expressing Hsp70 [48] induced a significant CD8+ T cell response, via enhanced antigen-specific T cell priming. Encapsulation of OC antigens in microparticles also increased CD4+ and CD8+ T cell populations in the ID8 model following immunisation [49]. In phase I clinical trials, immunization with a p53 peptide-based vaccine increased CD4+ and CD8+ T cells and improved progression-free survival rates in patients who developed p53-reactive T cells [50]. Nevertheless, no vaccine or targeted immune therapies indicated for OC have been successfully translated to the clinic to date. Our new ID8-iRFP720 model allows for the pre-clinical assessment of immune profiles in response not only to disease, but also to treatment, and can provide a direct, parallel visualization of tumour regression in response to therapy. This is an essential component of vaccine-based studies, where precise knowledge of timing for required vaccine prime/boost or the efficacy of specific adjuvants would be highly beneficial. Moreover, the efficacy of vaccination over time, ability to modulate specific immune populations, and direct determination of recurrence and influence of re-challenge following treatment can all be assessed. This will be crucial for the ongoing development of vaccine-based therapies that can translate successfully for OC treatment.

## 4. Materials and Methods

### 4.1. Plasmids

pROSA26-1 (*#*21714) [51], piRFP720-N1 (*#*45461) [21] and plentiCRISPRv1 (*#*49535) were purchased from Addgene (Watertown, MA, USA). pAAVS1-puro-DNR CRISPR vector (*#*GE100024) [52] was purchased from Origene (Rockville, MD, USA). All plasmids were transformed into competent *Escherichia coli STBL3*^TM^ cells (ThermoFisher Scientific, Franklin, MA, USA) and stored as glycerol stocks at −80 °C. *ROSA26* sequence guide strands were adapted from Quadros and colleagues [25] for cloning into lentiCRISPRv1 [52]. Guide strands used were ROSA26-sgRNA2 F: 5′-CACCGGTGTGTGGGCGTTGTCCTGC-3′; ROSA26-sgRNA2 R: 5′-AAACGCAGGACAACGCCCACACACC-3′; ROSA26-sgRNA4 F: 5′-CACCGATGTCTTTAATCTACCTCGA-3′; and ROSA26-sgRNA4 R: 5′-AAACTCGAGGTAGATTAAAGACATC-3′.

### 4.2. pROSA-puro-iRFP720 Vector Generation

pROSA26-1 was digested with *SbfI/XhoI* (NEB, Ipswich, MA, USA) to release a 4434 bp product. Digest products were separated by electrophoresis in a 1.2% agarose gel, excised and purified (Promega, Madison WI, USA). A QuikChange II XL Site-Directed Mutagenesis Kit (Agilent Technologies, Santa Clara, CA, USA) was used to introduce an *SbfI* restriction site at position 6849 (downstream from the polyA term site), and an *SgrDI* restriction site at position 4320 (3′ end of the puromycin resistance cassette) of the pAAVS1-puro-DNR vector. pAAVS1-puro-DNR was digested with *SbfI/SgrDI* (ThermoFisher Scientific) and ligated with the 4434bp *SbfI/XhoI* pROSA26-1 fragment to form the pAAVS1-ROSA26 vector. pAAVS1-ROSA26 was digested with *SgfI/MluI* (NEB) to release a 7054bp product which was purified by electrophoresis as above. A complete iRFP720 cassette was amplified by PCR from piRFP720-N1 using a Q5 high-fidelity polymerase and primers iRFP720-AsiSI F: 5′-TTTTGCGATCGCCACCATGGCG-3′ and iRFP720-MluI R: 5′-TTTTACGCGTGCCGCTCACTCTTCCA-3′. The amplified iRFP720 fragment was *AsiSI/MluI*-digested to release a 964bp product, which was ligated into pAAVS1-ROSA26 to generate the final pROSA-puro-iRFP720 vector (8018bp). The pROSA-puro-iRFP720 vector and accompanying ROSA26-sgRNA constructs for genomic integration are available from Ximbio (https://ximbio.com/search?q=pROSA&tab=products).

### 4.3. ID8 Cell Transfection and Cell Sorting

The ID8 mouse epithelial OC cell line (a gift from Dr. Kathy Roby, Kansas University Medical Center, Kansas City, KA, USA) was maintained in Gibco DMEM (ThermoFisher Scientific) containing 4% fetal bovine serum (FBS) with 1% insulin-transferrin-selenite (ITS) and 1% penicillin/streptomycin (PS). ID8 cells were grown to 70–80% confluence and then co-transfected with pROSA-puro-iRFP720 and plentiCRISPRv1 ROSA26-sgRNA2 or ROSA26-sgRNA4 using Lipofectamine 2000 (Invitrogen, CA, USA), using 3.5 µg of DNA in total. Transfected ID8 cells were maintained in puromycin selection for two weeks and then sorted by flow cytometry for single iRFP720+ clones ([R]730/45 vs. [R]670/30) using the FACSAria Fusion cell sorter (BD Biosciences, San Jose, CA, USA). Clones were maintained under selective pressure in DMEM/FBS/ITS/PS until outgrowth was evident.

### 4.4. Genomic Screening

Genomic DNA was isolated from iRFP720+ single-cell clones using the ISOLATE II Genomic DNA Kit (Bioline, Memphis, TN, USA) as described by the manufacturer. DNA was PCR-amplified using the following primers: *ROSA26* upstream F: 5′-GGCGTGTTTTGGTTGGCGTAAG-3′ (targeted to a genomic region upstream of the *ROSA26* left arm); *ROSA26* downstream R: 5′-ACCAGGTTAGCCTTTAAGCCTGC-3′ (targeted to an endogenous region between the *ROSA26* left and right arms); and iRFP720 R: 5′-TCGTCGCAGGTCAAGAGGTCA-3′. The ROSA26 upstream F and ROSA26 downstream R primers amplify an 869 bp band if the endogenous ROSA26 sequence has been retained, and the ROSA26 upstream F and iRFP720 R primers amplify a 2883 bp product upon successful incorporation of iRFP720 into *ROSA26.*

### 4.5. xCELLigence Assay

xCELLigence real-time cell analysis (ACEA Biosciences, San Diego, CA, USA) was used to assess adhesion and proliferation of wild-type and pROSA-iRFP720 ID8 cells as described previously [53]. Briefly, cells were cultured to 70–80% confluence and serum-starved overnight for cell-cycle synchronization. Cells were harvested with trypsin/EDTA, seeded into an E-Plate® 96 plate (ACEA Biosciences) at a concentration of 8 × 10^3^ cells/well, and allowed to adhere and proliferate for 8 h. After 8 h, cells were treated with cisplatin (10 µg/mL) or paclitaxel (20 nM) for further 64 h. Data are represented as CI, where CI = electrical impedance at timepoint *n*. 

### 4.6. Apoptosis Assay

Annexin V and propidium iodide (PI) staining was used to assess apoptosis by flow cytometry. Wild-type and pROSA-iRFP720 ID8 cells were seeded into a 6-well plate at a concentration of 3 × 10^5^ cells/well and were allowed to adhere overnight. Cells were treated with cisplatin (10 µg/mL) or paclitaxel (20 nM) and collected after 24 h by trypsinisation. Cells were stained with Alexa Fluor ® 647 Annexin V (BioLegend, San Diego, CA, USA) and propidium iodide (ThermoFisher Scientific) in annexin V binding buffer (10 mM HEPES, 140 mM NaCl, 2.5 mM CaCl_2_) for 15 min and analysed using the BD LSRFortessa™ X-20 (BD Biosciences). Unstained and single-stained cells were used to set compensation and gates.

### 4.7. Mice

Female 8-week-old C57BL/6 mice were obtained from Monash Animal Services (Clayton, VIC, Australia) and housed in a specific-pathogen-free (SPF) facility. All animal protocols were approved by the Alfred Medical Research and Education Precinct (AMREP) animal ethics committee, Melbourne, Australia (approval *#*E/1682/2016/M). Treatment and care of the animals were in accordance with institutional guidelines and with the Australian code for the care and use of animals for scientific purposes.

### 4.8. Intrabursal Implantation of ID8 pROSA-iRFP720 Tumours

Tumours were established by ovarian intrabursal (IB) implantation of ID8 cells as described previously [54]. Mice were anesthetised in an induction chamber using 3% isofluorane in 2 L/min oxygen and then maintained at 2% isofluorane in 1 L/min oxygen using a rodent facemask. An incision was made at the mid-dorsal region of the skin and the peritoneal membrane was excised at the latero-dorsal point above the location of the right ovary. The ovarian fat pad was externalised and stabilised with a serrefine clamp. Either 1 × 10^6^ wild-type or pROSA-iRFP720 ID8 cells were loaded into a Hamilton microliter syringe (Sigma-Aldrich, St Louis, MO, USA) and injected underneath the ovarian bursa using a dissecting microscope for guidance. The skin was closed using Michel suture clips. Mice were monitored for recovery, and suture clips were removed 7 days post-surgery. Mice were fed an SF-AIN-93M rodent diet to reduce intrinsic autofluorescence. Mice were monitored weekly for weight and circumference, and humane endpoints were determined by a body abdominal circumference of >100 mm and general wellbeing (lack of responsiveness, hunched posture, ruffled fur, and eyes squinted). Once the endpoint was reached, mice were humanely sacrificed by CO_2_ asphyxiation, and blood and tissues were harvested for fluorescence imaging, ELISA, flow cytometry, and formalin fixed for immunofluorescence.

### 4.9. In Vivo Fluorescence Imaging

In vivo near-infrared fluorescence imaging was performed weekly using an IVIS Lumina III In Vivo Imaging System (PerkinElmer, Boston, MA, USA). Wild-type or pROSA-iRFP720 ID8 tumour-bearing mice were anesthetised using 3% isofluorane in 2 L/min oxygen, a small patch of abdominal fur shaved, and placed into the IVIS Lumina under 2% isofluorane in 1 L/min oxygen. Mice were imaged at field of view (FOV) C with a brightfield and X-ray set on auto-exposure, and iRFP720 fluorescence was detected using the spectral unmixing iRFP filter set (excitation: 620 nm, 640 nm, 660 nm, and 680 nm; emission: 710 nm), at an exposure time of 5 s. Spectral unmixing of iRFP720 signal from intrinsic autofluorescence was performed by subtracting 680_ex_/710_em_ fluorescence values from the other channels, and a custom spectral unmixing algorithm was created using known iRFP720 fluorescence from a pROSA-iRFP720 ID8 mouse vs. autofluorescence in a wild-type ID8 mouse. For culled animals, the ovaries, omentum, liver, kidneys, small intestine and spleen were excised post-mortem and imaged using the IVIS Lumina. All quantitative fluorescence measurements and analyses were performed using the Living Image software (v 4.5.1, PerkinElmer).

### 4.10. Tissue Immunofluorescent Staining

Immunofluorescent staining was performed on 4 µm formalin-fixed, paraffin-embedded ovarian tumour sections. After deparaffinization, antigen retrieval was performed by boiling slides in sodium citrate buffer, and permeabilization was performed for nuclear antigens only using 0.25% *v*/*v* Triton X-100. Non-specific antibody binding was blocked with 5% BSA in PBS, and primary antibodies were applied and incubated overnight at 4 °C. Primary antibodies used were rabbit anti-wide-spectrum cytokeratin (*#*ab9377, 1:100, Abcam, Cambridge, UK), rabbit anti-Wilms tumor protein-1 (*#*ab89901, 1:100, Abcam), goat anti-PD-L1 (*#*AF1019, 1:125, R&D Systems, Minneapolis, MN, USA) and rabbit anti-p53 (*#*ab31333, 1:100, Abcam). Alexa Fluor® 594 or 647-conjugated goat anti-rabbit or donkey anti-goat secondary antibodies (1:1000, Abcam) were applied at 1:1000 for 1 h, and slides were mounted using ProLong™ Gold Antifade Mountant with DAPI (ThermoFisher Scientific). Fluorescence images were captured using the Cytation 3 imaging Multi-Mode Reader (BioTek) and processed using the Gen5™ software v 2.05 (BioTek).

### 4.11. Enzyme-linked Immunosorbent Assay

The presence of iRFP720-specific IgG antibodies in sera was assessed by ELISA as described [55]. iRFP720 protein was purified from *E. coli* cells using a Ni-NTA Fast Start kit (Qiagen, Hilden, Germany). Standards were mouse IgG (Sigma-Aldrich, MO, USA) from 0.5 µg/mL to 0.00781 µg/mL. All measurements were carried out in triplicate. The detection antibody was an Alexa-594-conjugated anti-mouse IgG (5 µg/mL, Abcam). Fluorescence was measured using a Cytation 3 MultiMode Plate Imager (BioTek), with fluorescence (590 nm excitation, 617 nm emission) acquired using a fixed probe height of 6.75 mm, 200 data-points per well and dynamic range 100–80,000 units. Standard curves were constructed using Gen5.0 software v 2.05 (BioTek). A positive IgG serum titre was defined as a reading of >2 standard deviations above the mean for mice implanted with wild-type ID8 cells.

### 4.12. Flow Cytometry

Leukocytes from lymph nodes, blood, spleen and peritoneal cavity wash were collected and isolated as described [56], and analysed by flow cytometry. Anti-mouse CD16/CD32 antibody (1:50, BD Biosciences, clone #2.4G2) was used to block non-specific binding. The following fluorochrome-conjugated antibodies were used: CD3e-AF700 (1:200, BD Biosciences, clone #500A2), CD4-BUV395 (1:100, BD Biosciences, clone #RM4-5), CD8a-PerCP (1:100, BioLegend, clone #53-6.7), CD25-PE-CF594 (1:100, BD Biosciences, #PC-61), FoxP3-APC (1:50, eBioscience, clone #FJK-16S), CD11b-APC (1:200, BioLegend, clone #MI/71), CD11c-BV421 (1:50, BD Bioscience, clone #HL3), GR1-PerCP Cy5.5 (1:200, BD Biosciences, clone #RB6-8C5), F4/80-BV711 (1:100, BioLegend, clone #BM8) and MHCII-APC/Cy7 (1:500, BioLegend, clone #M5/114.15.2). Single-stained beads were used to set compensation controls, and fluorescence-minus-one (FMO) controls were used to define population gates. Data were acquired using a BD LSRFortessa X-20 (BD Biosciences), and were analysed using FlowJo software v10.5.0 (BD Biosciences).

### 4.13 Statistical Analysis

Graphs and statistical analyses were performed using GraphPad Prism v 7.0b (GraphPad Software Inc., La Jolla, CA, USA). A two-way ANOVA and Sidak multiple comparisons test were used to analyse statistical significance between groups for apoptosis, iRFP720 fluorescence post-mortem, and immune populations. A two-way ANOVA and Tukey post-hoc test was used to analyse statistical significance between groups for proliferation, iRFP720 fluorescence, weight and circumference over time. A non-parametric one-way ANOVA and Tukey post-hoc test was used for the IgG assay. Data are presented as mean ± SD and *p* < 0.05 was considered statistically significant. 

## 5. Conclusions

Our data demonstrate that the pROSA-iRFP720 ID8 model is a suitable tool for non-invasive, longitudinal tumour monitoring in vivo, and provides the first simple fluorescence model for real-time discrimination between distinct stages of ovarian tumour progression. All specific features of host–tumour immune interactions that parallel human clinical progression are preserved in this system, providing a tool for the accurate preclinical development, evaluation and testing of novel vaccine and immune-based therapies for OC. By directly assessing tumour regression, recurrence and response to therapies at an early stage, it will be possible to accurately determine therapeutic scheduling and optimize combination therapies to achieve clinically relevant and robust outcomes for OC treatment.

## Figures and Tables

**Figure 1 cancers-11-00032-f001:**
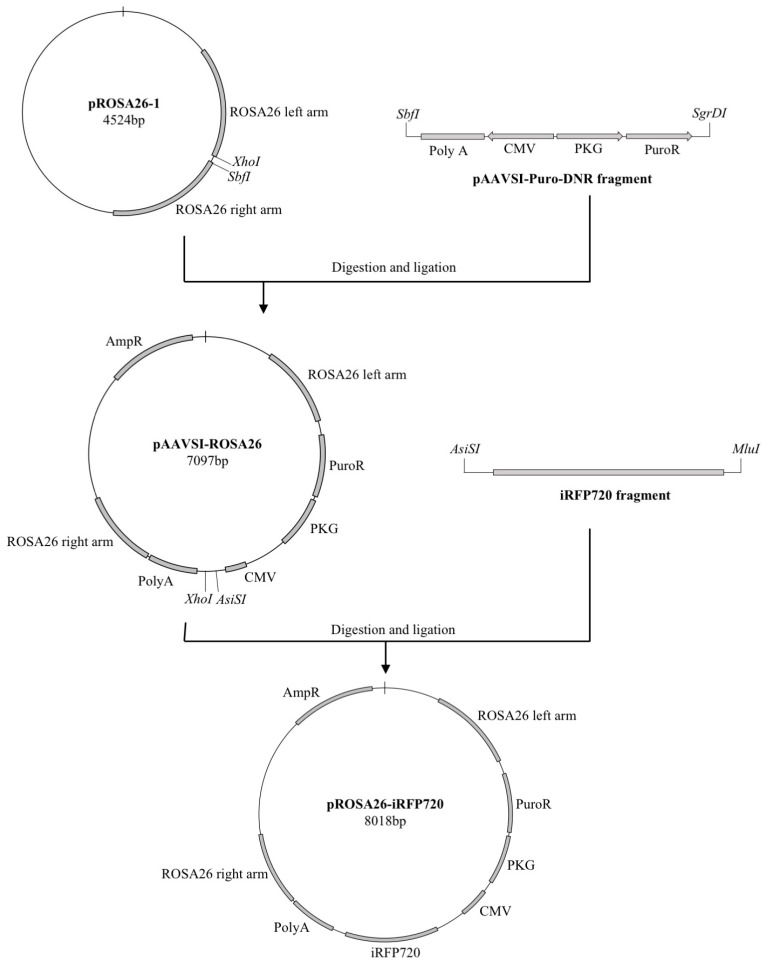
Strategy for pROSA-puro-iRFP720 vector generation and genomic screening. Schematic representation of the overall cloning strategy for the generation of the pROSA-puro-iRFP720 vector. A fragment from the pAAVS1-puro-DNR plasmid was PCR-amplified, digested with SbfI and SgrDI and ligated into an SbfI/XhoI-digested pROS A26-1 plasmid. An iRFP720 fragment from piRFP720-N1 was PCR-amplified, digested with AsiSI and MluI and ligated into the AsiSI/MluI-digested pAAVSI-ROSA26 vector.

**Figure 2 cancers-11-00032-f002:**
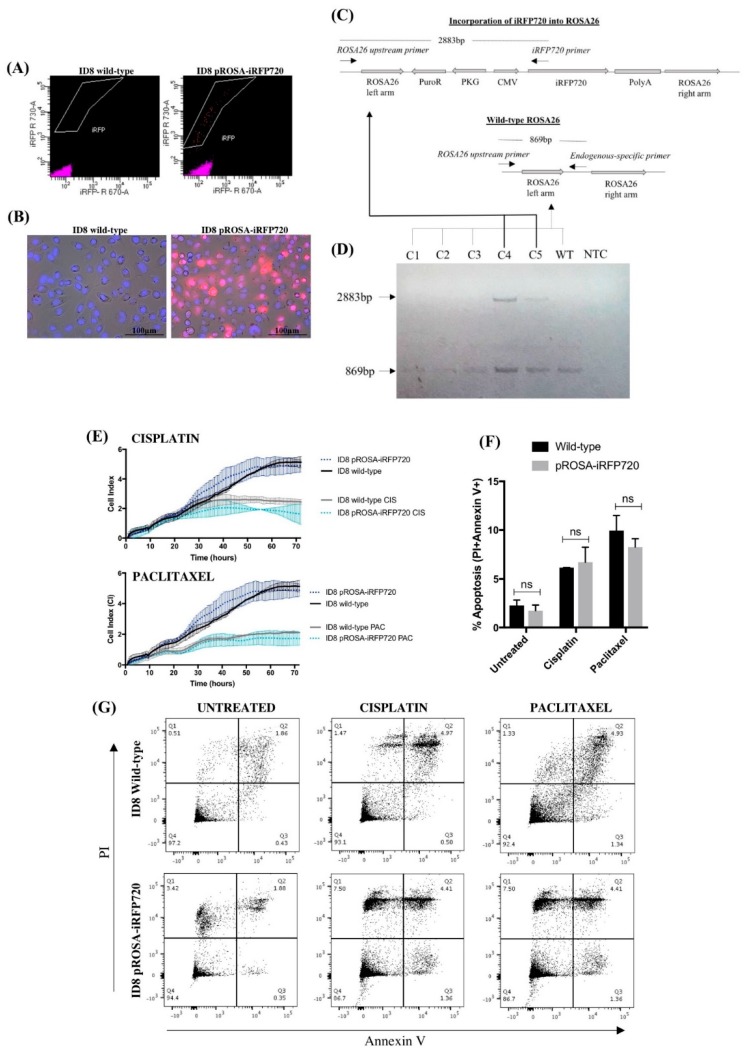
iRFP720 fluorescence and characterisation of pROSA-iRFP720-expressing ID8 cells. (**A**) Fluorescence-activated cell sorting (FACS) enrichment of iRFP720-expressing ID8 cells. iRFP720-positive single cells ([R]730/45 vs. [R]670/30) were sorted into a 96-well plate using the FACSAria Fusion cell sorter (BD Biosciences, San Jose, CA, USA), and single clones were expanded and screened. (**B**) iRFP720 fluorescence was observed using the Cytation 3 imaging Multi-Mode Reader equipped with a Cy5.5 filterset (BioTek Instruments Inc., Winooski, VT, USA). Hoechst 33342 was used to visualise the nucleus (377_ex_/447_em_). (**C**) Schematic diagram representing the genomic screening strategy used for identifying iRFP720 incorporation into the *ROSA26* locus. (**D**) Genomic DNA isolated from pROSA-puro-iRFP720-transfected single-cell ID8 clones was PCR-amplified using a *ROSA26* upstream F primer, an endogenous *ROSA26*-specific R primer and an iRFP720-specific R primer, and PCR products were separated with a 1.2% agarose gel. Wild-type *ROSA26* gives an 869bp product and successful iRFP720 incorporation gives a 2883bp product. C1–5: clones 1–5; WT: wild-type ID8; NTC: no-template control. (**E**) Proliferation was assessed by electrode impedance using xCELLigence real-time cell analysis. Cells (8 × 10^3^/well) were treated with cisplatin (10 µg/mL) or paclitaxel (20 nM) after 8 h, and proliferation was analysed for further 64 h. (**F**) Wild-type and pROSA-iRFP720 ID8 (3 × 10^5^ cells/well) were cultured for 24 h, and then incubated for further 24 h in the presence of cisplatin (10 µg/mL) or paclitaxel (20 nM). Apoptosis was determined by Alexa Fluor®647 annexin V and PI staining on the BD LSRFortessa X-20 (BD Biosciences) flow cytometer. (**G**) Representative images of the apoptosis analysis. Data are presented as mean ± SD, *n* = 3.

**Figure 3 cancers-11-00032-f003:**
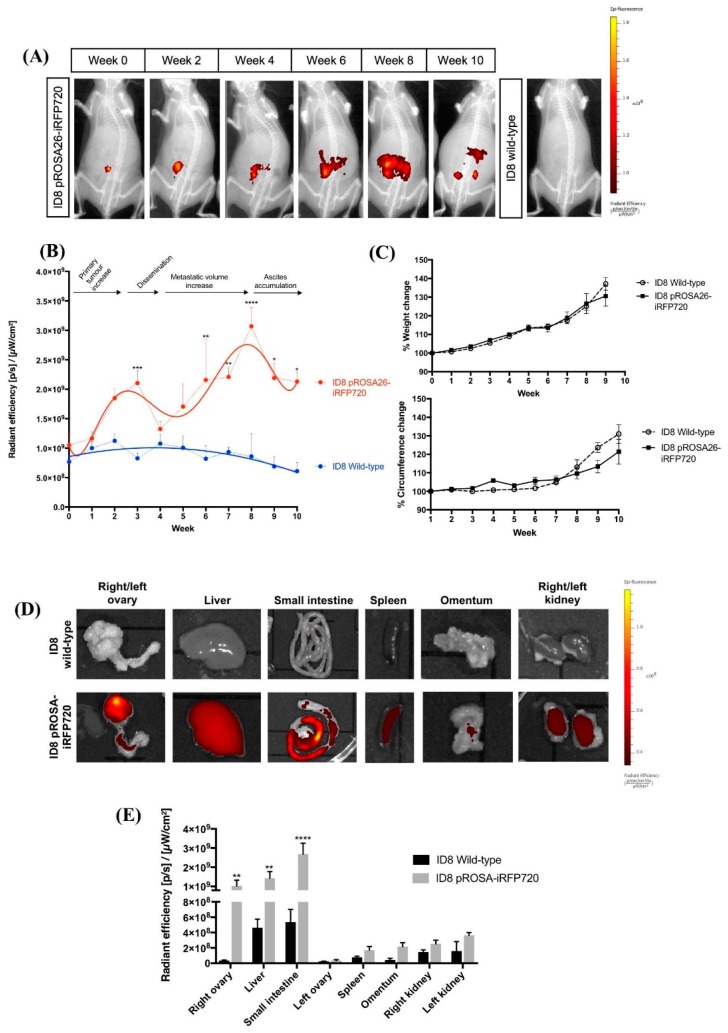
In vivo tracking of iRFP720 fluorescence in real time. (**A**) pROSA-iRFP720 or wild-type ID8 cells (1 × 10^6^ cells/mouse) were implanted into the ovarian bursa and iRFP720 fluorescence was measured using the IVIS Lumina III In Vivo Imaging System (Perkin Elmer, MA, USA). Mice were imaged at field of view (FOV) C (*n* = 4–16) and iRFP720 fluorescence was detected using the iRFP filter set at an exposure time of 5 s. The iRFP signal was isolated using a custom spectral unmixing algorithm. Representative images are shown. (**B**) Quantitative region of interest (ROI) analysis of iRFP total radiant efficiency (p/s)/(µW/cm^2^) over time. (**C**) Average weight and circumference of mice over time following tumour implantation (*n* = 4–16). (**D**) Representative images of iRFP720 fluorescence in excised organs post-mortem at the humane endpoint. (**E**) Quantitative ROI analysis of excised organs indicating iRFP total radiant efficiency (p/s)/ (µW/cm^2^) (*n* = 4,5). *, *p* < 0.05; **, *p* < 0.01; ***, *p* < 0.001; ****, *p* < 0.0001.

**Figure 4 cancers-11-00032-f004:**
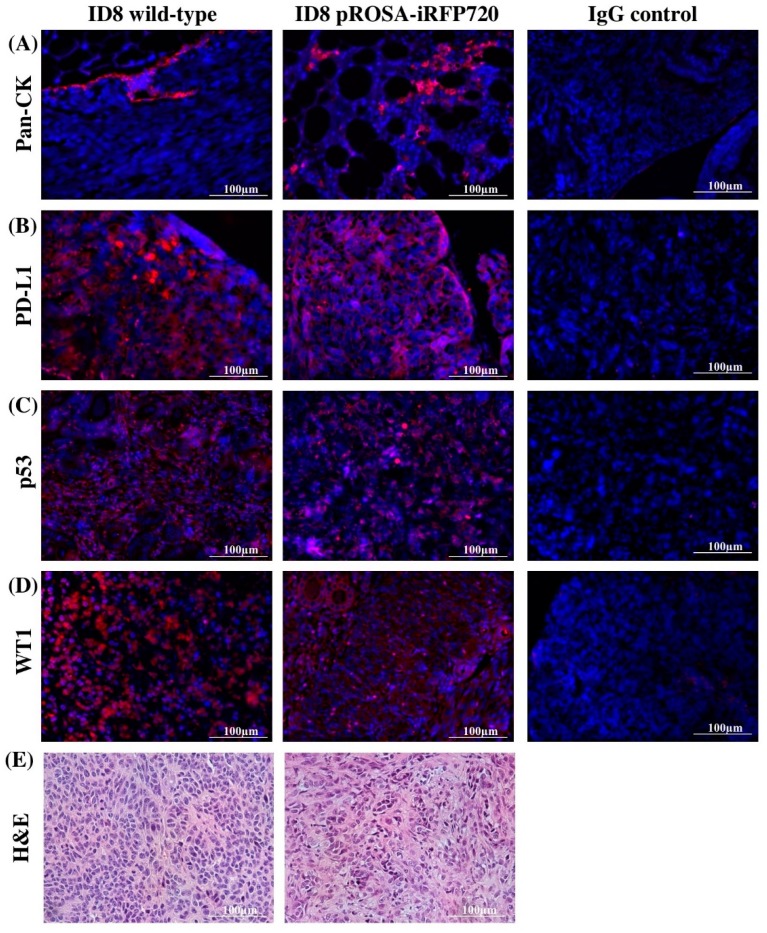
Expression of serous tumour markers by immunofluorescence. Representative fluorescent images of pROSA-iRFP720 and wild-type ID8 ovarian tumour sections stained with (**A**) pan-cytokeratin (pan-CK), (**B**) PD-L1, (**C**) p53 and (**D**) Wilms Tumour protein 1 (WT1). Images were acquired using the Cytation 3 imaging Multi-Mode Reader (BioTek) and were processed using the Gen5™ software v 2.05 (BioTek). Staining appears red against nuclei stained with DAPI (blue). (**E**) Wild-type and pROSA-iRFP720 ID8 ovarian tumour sections stained with hematoxylin and eosin (H&E).

**Figure 5 cancers-11-00032-f005:**
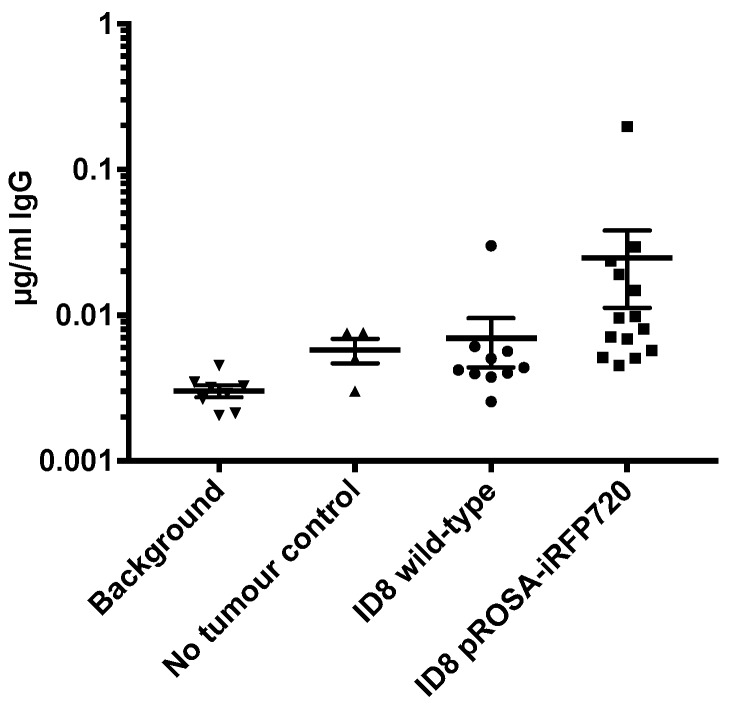
Circulating endogenous iRFP720-specific IgG antibodies in mice bearing pROSA-iRFP720 ID8 tumours. Serum iRFP720-specific IgG antibodies (µg/mL) in the sera of pROSA-iRFP720 ID8 and wild-type ID8 tumour-bearing mice at the endpoint (*n* = 4–14). Bars represent ± SD.

**Figure 6 cancers-11-00032-f006:**
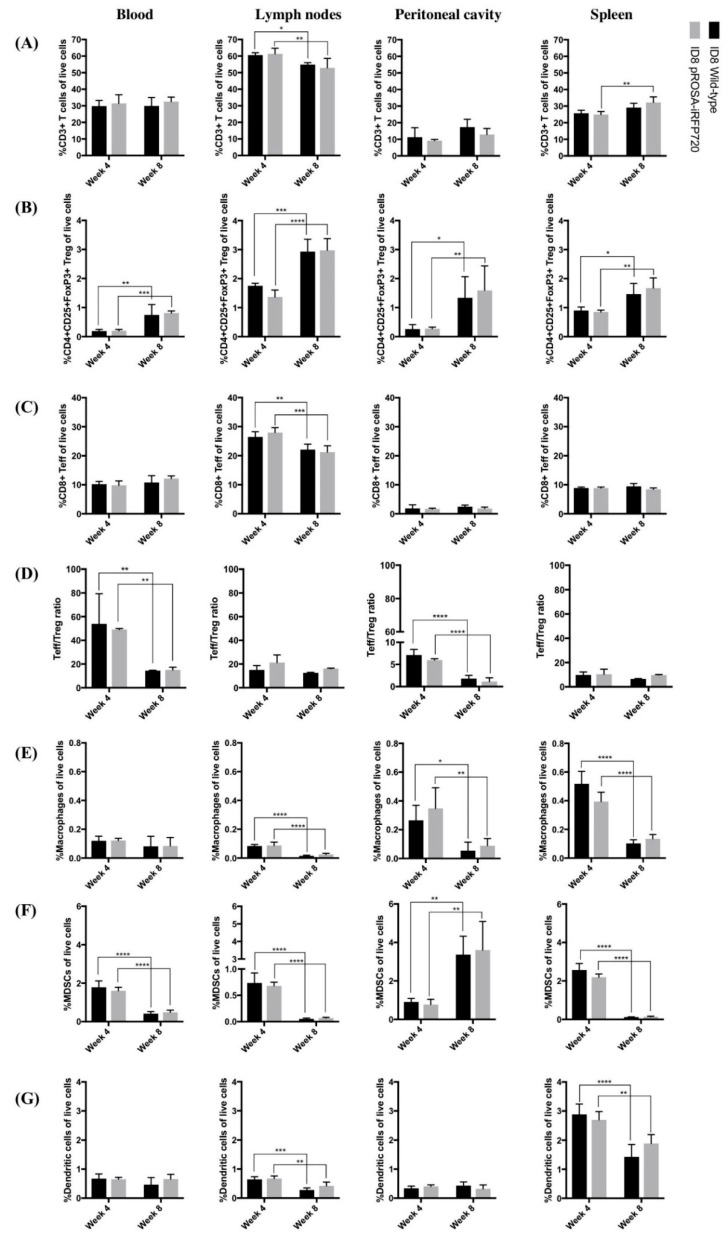
Changes in immune populations over time in mice bearing pROSA-iRFP720 ID8 tumours. Percentages of (**A**) total CD3+ T cells, (**B**) T regulatory cells (CD3+CD4+CD25+FoxP3+), (**C**) CD8+ T effector cells (CD3+CD8+) of live cells; (**D**) Teff/Treg ratios; percentages of (**E**) macrophages (CD3−CD11b+F4/80+MHCII+), (**F**) myeloid-derived suppressor cells (MDSCs) (CD3–CD11c−CD11b+GR1+) and (**G**) dendritic cells (CD3−GR1−CD11c+) of live cells in the blood, lymph nodes, peritoneal cavity and spleen of pROSA-iRFP720 and wild-type ID8 tumour bearing mice at 4 weeks and 8 weeks post-inoculation. Data are presented as mean ± SD, *n* = 4,5. *, *p* < 0.05; **, *p* < 0.01; ***, *p* < 0.001; ****, *p* < 0.0001.

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
