# Peer review of "Non-Invasive Fluorescent Monitoring of Ovarian Cancer in an Immunocompetent Mouse Model"

_cancers, 2018, doi:10.3390/cancers11010032_

Round 1

Reviewer 1 Report

The authors present a new model which permits non-invasive longitudinal tumour monitoring whilst preverving host-tumour immune interactions and allows the pre-clinical assessment of immune profiles throughout disease progression as well as the direct visualization of therapeutic response. 

Immune-mediated therapies will be a new challenge for the treatment of ovarian cancers. This paper can open the discussion on possibility to monitor tumour regression, recurrence and response to therapies at an early stage of ovarian cancer in an immunicompetent mouse model.

In this context, I think that this paper is well written and it can be for publication.

I underline that Materials and Methods have been well conducted and reported exhaustively.

I  invite the authors for:

Minor revisions:

Introduction: I think that the introduction paragraph is too long and I ask to reduce it

Pag.9:  paraghaph 2.5: Authors report that only 2/14 mice had a detectibile titre of anti-iRFP720 IgG and they report that there was not discernibile effect on disease progression in either of these animals or for iRFP+ID8 tumour mice generally compared to wild type. Please I invite authors to report this consideration in the discussion, reporting their sentence that iRFP720 was only weakly immunogenic in mice and its expression did not alter significantly tumour progression in vivo

Author Response

Dear Reviewer 1, 

Thank you for reviewing our manuscript. Please find specific responses to your questions below.

R1-1: Introduction: I think that the introduction paragraph is too long and I ask to reduce it

We have made changes to the introductory section, and reduced its length substantially as requested.

R1-2: Pag.9:  paraghaph 2.5: Authors report that only 2/14 mice had a detectibile titre of anti-iRFP720 IgG and they report that there was not discernibile effect on disease progression in either of these animals or for iRFP+ID8 tumour mice generally compared to wild type. Please I invite authors to report this consideration in the discussion, reporting their sentence that iRFP720 was only weakly immunogenic in mice and its expression did not alter significantly tumour progression in vivo

We have added further text to the discussion to help clarify this point. The relevant text now reads, “Integration of a constitutively expressed iRFP720 at the ROSA26 genomic locus [37] had no discernible consequences for cell growth in vitro, nor did it induce any significant changes in either cellular or humoral immune responses in vivo. Moreover, low titre anti-iRFP antibodies could only be detected in 2/14 mice with advanced disease; and iRFP expression had no discernible impact on tumour growth, progression or survival time in any animals”.

Reviewer 2 Report

The subject matter of this manuscript is topical and some reported data are indeed new and interesting. In my opinion, however, the article needs minor improvements:

P.2, l. 91: Quotation [22] should be supplemented by additional quotation(s), e.g. Morvova M.,Jr.; Jeczko P.; Sikurova L. Gender differences in the fluorescence of human skin in young healthy adults. Skin Research and Technology 2018, 24, 599-605  

P.2, l.  93 - 94: The sentence “Near-infrared light passes through tissue more easily, due to combined absorption of melanin, haemoglobin and water.” is problematic. I think it is necessary to properly re-formulate it and quote it properly.

P.5, l. 146: Please justify the use of 647 nm (excitation) and 794 nm (emission) for iRFP720. I would appreciate the demonstration of the iRFP720 spectra. Please discuss adequacy of the iRFP720 fluorescence signal (yield) for the given wavelengths.

Author Response

Dear Reviewer 2,

Thank you for reviewing our manuscript, and for your suggestions. Please find below responses to your questions.

R2-1:  P.2, l. 91: Quotation [22] should be supplemented by additional quotation(s), e.g. Morvova M.,Jr.; Jeczko P.; Sikurova L. Gender differences in the fluorescence of human skin in young healthy adults. Skin Research and Technology 2018, 24, 599-605

This reference has been added as suggested.

R2-2: P.2, l.  93 - 94: The sentence “Near-infrared light passes through tissue more easily, due to combined absorption of melanin, haemoglobin and water.” is problematic. I think it is necessary to properly re-formulate it and quote it properly.

 We apologize for this ambiguous statement. The sentence has been changed to read, “Tissue absorption in the near infra-red spectrum is minimal, and the relatively low level of autofluorescence and light scatter by mammalian tissues at these wavelengths make iRFPs desirable for deep tissue imaging [21]”.

R2-3: P.5, l. 146: Please justify the use of 647 nm (excitation) and 794 nm (emission) for iRFP720. I would appreciate the demonstration of the iRFP720 spectra. Please discuss adequacy of the iRFP720 fluorescence signal (yield) for the given wavelengths.

We apologize for this incorrect statement in the legend of Figure 2. Cellular iRFP720 fluorescence was visualized using a standard Cy5.5 filterset, with fluorescence emission detected within the range 695 to 796nm. The text has been amended to read, “iRFP720 fluorescence was observed using the Cytation 3 imaging Multi-Mode Reader equipped with a Cy5.5 filterset (BioTek, VT, USA)”.